# Clinical Application of Next-Generation Sequencing of Plasma Cell-Free DNA for Genotyping Untreated Advanced Non-Small Cell Lung Cancer

**DOI:** 10.3390/cancers13112707

**Published:** 2021-05-30

**Authors:** Maria Gabriela O. Fernandes, Natália Cruz-Martins, Conceição Souto Moura, Susana Guimarães, Joana Pereira Reis, Ana Justino, Maria João Pina, Adriana Magalhães, Henrique Queiroga, José Carlos Machado, Venceslau Hespanhol, José Luis Costa

**Affiliations:** 1Pulmonology Department, Centro Hospitalar Universitário de São João, Alameda Prof. Hernani Monteiro, 4200-319 Porto, Portugal; maria.adriana.magalhaes@chsj.min-saude.pt (A.M.); hqueiroga@chsj.min-saude.pt (H.Q.); vhespanhol@chsj.min-saude.pt (V.H.); 2Faculty of Medicine, University of Porto, Alameda Prof. Hernani Monteiro, 4200-319 Porto, Portugal; ncmartins@med.up.pt (N.C.-M.); susana.guimaraes@chsj.min-saude.pt (S.G.); josem@ipatimup.pt (J.C.M.); jcosta@ipatimup.pt (J.L.C.); 3Institute of Molecular Pathology and Immunology of the University of Porto (IPATIMUP), 4200-135 Porto, Portugal; joanar@ipatimup.pt (J.P.R.); ajustino@ipatimup.pt (A.J.); mpina@ipatimup.pt (M.J.P.); 4Institute for Research and Innovation in Health (i3S), University of Porto, Rua Alfredo Allen, 4200-135 Porto, Portugal; 5Laboratory of Neuropsychophysiology, Faculty of Psychology and Education Sciences, University of Porto, 4200-135 Porto, Portugal; 6Pathology Department, Centro Hospitalar Universitário de São João, Alameda Prof. Hernani Monteiro, 4200-319 Porto, Portugal; m.moura@chsj.min-saude.pt

**Keywords:** circulating tumor DNA (ctDNA), cell-free DNA (cfDNA), next generation sequencing (NGS), lung adenocarcinoma, liquid biopsy, genotyping

## Abstract

**Simple Summary:**

Plasma ctDNA is a material source for molecular analysis particularly useful when tissue is not available or sufficient. NGS-based plasma genotyping should be integrated into the clinical workup of newly diagnosed advanced NSCLC.

**Abstract:**

Background: Analysis of circulating tumor DNA (ctDNA) has remarkable potential as a non-invasive lung cancer molecular diagnostic method. This prospective study addressed the clinical value of a targeted-gene amplicon-based plasma next-generation sequencing (NGS) assay to detect actionable mutations in ctDNA in patients with newly diagnosed advanced lung adenocarcinoma. Methods: ctDNA test performance and concordance with tissue NGS were determined, and the correlation between ctDNA findings, clinical features, and clinical outcomes was evaluated in 115 patients with paired plasma and tissue samples. Results: Targeted-gene NGS-based ctDNA and NGS-based tissue analysis detected 54 and 63 genomic alterations, respectively; 11 patients presented co-mutations, totalizing 66 hotspot mutations detected, 51 on both tissue and plasma, 12 exclusively on tissue, and 3 exclusively on plasma. NGS-based ctDNA revealed a diagnostic performance with 81.0% sensitivity, 95.3% specificity, 94.4% PPV, 83.6% NPV, test accuracy of 88.2%, and Cohen’s Kappa 0.764. PFS and OS assessed by both assays did not significantly differ. Detection of ctDNA alterations was statistically associated with metastatic disease (*p* = 0.013), extra-thoracic metastasis (*p* = 0.004) and the number of organs involved (*p* = 0.010). Conclusions: This study highlights the potential use of ctDNA for mutation detection in newly diagnosed NSCLC patients due to its high accuracy and correlation with clinical outcomes.

## 1. Introduction

Lung cancer (LC) biomarker-driven therapy changed the treatment paradigm of advanced NSCLC. Identification of several oncogenic driver mutations and the approval of an increasing number of targeted drugs over the past years contributed to improved LC patients’ survival. Tumor genotyping is mandatory for selecting patients suitable for targeted treatments and is advised for all patients with advanced non-squamous NSCLC and some selected patients with squamous cell carcinoma, according to the current guidelines [1,2,3]. Additionally, at the time of disease progression, the identification of molecular resistance mechanisms is extremely relevant, as targetable alterations can be found. In addition to molecular testing, immunohistochemical PD-L1 is required for all NSCLC patients to look for candidates for checkpoint inhibitors [4].

However, despite these guidelines, under-genotyping is still a problem. Real-world data reveals that the implementation of molecular testing is heterogeneous. In a significant proportion of patients, minimum biomarker testing guidelines is not achieved [5,6]. Beyond accessibility and economic factors, one reason for insufficient molecular testing is tumor sample exhaustion. Insufficient tissue for genotyping is common among centers, ranging from 5% to 25% [5,6]. In our center, before NGS implementation, ALK testing after EGFR was not performed in 18% of the patients due to lack of sample; however, with a combined DNA and RNA targeted-gene amplicon NGS based panel, the rate of molecularly unclassifiable patients reduced from 73% to 36% [7].

Tissue biopsy is the gold standard for detecting oncogenic mutations; however, in LC, tumor samples are usually obtained by invasive methods and often have scarce tumor content, configuring the adoption of new technologies, such as NGS and circulating tumor DNA (ctDNA) assays, an option to help matching patients to targeted therapies [8]. Circulating tumor DNA (ctDNA) is a small part of cell-free DNA (cfDNA) present in plasma and represents the DNA released mainly by apoptosis and necrosis of tumor cells and corresponding metastasis [9]. The genomic profiling of ctDNA provides a non-invasive alternative to tissue biopsies and gives a broader image of tumor heterogeneity being currently considered a liquid biopsy.

For plasma ctDNA analysis, NGS is becoming widely available, despite the lack of standardization and recommendations for its use. Indeed, NGS allows the sequencing of several genomic regions in a single test, on a single platform and in samples with low tumor DNA content, such as plasma. The presence of somatic mutations in the plasma cfDNA proved to be a surrogate marker for response to targeted therapies [10] and resistance mechanisms [11]. In the setting of EGFR-mutated tumors progressing under first or second-generation EGFR tyrosine-kinases, detecting the T790M mutation is already approved in ctDNA [12]. Nonetheless, the ctDNA analysis goes beyond the detection of resistance mechanisms.

The American Society and Clinical Oncology and the College of American Pathologists do not recommend the use of cfDNA due to insufficient validity [13]. However, a statement from IASCL considered that a liquid biopsy could be used at the time of initial diagnosis, being especially relevant when tumor tissue is scarce, unavailable, or when a tissue biopsy delay is expected [14]. The application of NGS to ctDNA can be a valuable tool for genotyping newly diagnosed patients with LC. In this sense, an amplicon-based technology was used to detect genomic alterations in ctDNA. Test performance was also calculated, and the determinants of plasma positivity and correlations with clinical outcomes addressed.

## 2. Materials and Methods

### 2.1. Study Design

Patients newly diagnosed with advanced lung adenocarcinoma (unresectable stage II and stage IV) at the Pulmonology Department of the Centro Hospitalar e Universitário de São João (CHUSJ), Porto—Portugal, were prospectively enrolled from 2015 until 2016. Data were censored for follow-up evaluation in May 2020, when 90% of deaths had occurred.

Of 127 patients, 115 had matched tumor and blood samples available for NGS testing at diagnosis. Tumor staging was based on the 8th edition from January 2018. The TNM staging of patients included until December 2017 was reclassified according to the 8th edition [12,13]. Tumor size (T) was measured by the longest diameter of the primary lesion assessed by CT scan and adenopathies (N) by the short axis by CT scan [15].

All subjects gave their informed consent for inclusion in the study. The study was conducted following the Helsinki Declaration, and the CHUSJ Ethics Committee approved the study protocol (CES-108/14). The study design is shown in Appendix A.

### 2.2. Study Outcomes

The test performance was calculated according to the standard definitions for sensitivity, specificity, negative (NPV) and positive predictive (PPV) values and accuracy. Concordance between tumor and plasma hotspot oncogenic alterations was determined. Digital PCR confirmed the discordant cases. Correlations between clinical features and plasma ctDNA results were investigated. Differences in progression-free survival (PFS) and overall survival (OS) between plasma and tissue NGS data were compared. PFS was defined as the time from the first-line treatment initiation to disease progression, death from any cause, or last follow-up date, and OS as the time from the diagnosis to death or last follow-up date.

### 2.3. Plasma and Tumor Tissue Genotyping

Whole-blood, plasma cell-free DNA isolation and library construction is detailed in the Appendix A. Blood samples were collected in K2EDTA BD Vacutainer^®^ PPT™ Plasma Preparation Tube (Becton Dickinson, Franklin Lakes, NJ, USA). DNA was extracted with the MagMax Cell-Free Total Nucleic Acid Isolation Kit (Thermo Fisher Scientific, Waltham, MA, USA) and quantified with the dsDNA HS assay kit by Qubit 3.0 or 4.0 Fluorometer (Thermo Fisher Scientific, Waltham, MA, USA). Targeted plasma NGS was performed using a validated amplicon-based NGS Oncomine™ Lung cfDNA Assay (Thermo Fisher Scientific, Waltham, MA, USA) that uses target gene enrichment by PCR with a set of primers for exons or hotspots of the selected gene, to detect single nucleotide variants (SNV) and short indels, covering more than 150 hotspots on *ALK, BRAF, EGFR, ERBB2, KRAS, MAP2K1, MET, NRAS, PIK3CA, ROS1* and *TP53*. Sequencing and bioinformatic analysis are detailed in Appendix A.

Biopsy and cytology specimens were reviewed by a pathologist. Histological specimens were fixed with formalin (formalin-fixed paraffin-embedded tissue, FFPE) and cytological specimens as smears or cellblocks. Samples were used for DNA extraction using the QIAamp DNA Mini Kit (Qiagen, Hilden, Germany), following the manufacturer’s instructions. To detect DNA changes, 4–5 FFPE tissue sections of 10 µm thickness with at least 10% tumor cells were used to isolate DNA. DNA was quantified with NanoDrop Lite Spectrophotometer (Thermo Fisher Scientific., Waltham, MA, USA) or Qubit^®^ 2.0 Fluorometer (Invitrogen, Waltham, MA, USA).

The Ion AmpliSeq Colon and Lung Cancer Research Panel v2 (Ion Torrent, Waltham, MA, USA) was used to detect DNA changes. This multiplex PCR-based test allows the analysis of 1850 hotspots and targeted regions in 22 genes (*AKT1, ALK, BRAF, CTNNB1, DDR2, EGFR, ERBB2, ERBB4, FBX7, FGFR3, FGFR1, FGFR2, KRAS, MAP2K1, MET, NOTCH1, NRAS, PTEN, PIK3CA, STK11, SMAD4,* and *TP53*). Libraries were generated using 1–10 ng of DNA from tissue FFPE blocks sections, according to the manufacturer. Sequencing and bioinformatic analysis are detailed in Appendix A. Digital PCR validated discordant mutations detected by both NGS assays (Appendix A).

The variant allelic fraction (VAF) was reported as the number of mutated DNA molecules divided by the total number of DNA fragments of that allele and is presented as a percentage.

Both tissue and plasma assays were performed at the Institute of Molecular Pathology and Immunology of the University of Porto—Portugal (IPATIMUP), a College of American Pathologists and ISO15189 accredited laboratory.

### 2.4. Statistical Analysis

The sample size was limited by the availability of specimens for both tumor and plasma NGS analysis. Most analyses were descriptive. Categorical data were described as absolute (n) and relative frequencies, while continuous variables were described as medians, interquartile ranges (IQR), and minimum and maximum values. NGS results were correlated with other parameters and assessed by the chi-square or Fisher’s exact tests, when appropriate. The unweighted Cohen’s kappa coefficient was used to assess the inter-rater agreement for categorical data. To analyze the ctDNA test performance, tissue DNA was considered the reference (True Positive: same alteration in tissue and plasma DNA; True Negative: no mutation in both samples; False Negative: mutation present in tissue but not in plasma; False Positive: mutation present in plasma but not in tissue). To have a deeper understanding of the ctDNA positivity-associated factors, multivariate logistic regression analysis was applied. Kaplan-Meier actuarial curve analysis was used to assess survival and the log-rank test for the chi-square calculus for each event time and each group considered. The significance level assumed was 0.05. All statistical analyses were performed using the Statistical Package for Social Sciences (SPSS, IBM Corp, Chicago, IL, USA) software, version 25.0.

## 3. Results

### 3.1. Patient’s Characteristics, Disease Extension, and Tumor Burden

The patient’s demographics and clinical characteristics are displayed in Table 1. Of the 115 patients included, with a median age of 66 (minimum: 38; maximum: 92) years, most were males (61.7%), smoker or former smokers (63.5%), and were at stage IV of the disease (81.7%). Samples were predominantly core biopsies suitable for histological analysis. Cytological samples were obtained by fine-needle-aspiration techniques, as endobronchial ultrasound-guided needle aspiration (EBUS-TBNA), lung and peripheral lymph nodes fine-needle aspirations, and pleural and pericardial fluid aspiration (Appendix A). Regarding TNM discriminators, 44.3% of patients were T4; 53.9% had N2/N3 disease; extra-thoracic involvement was found in 63.5% of patients, and 45.2% were at stage M1c with a median number of organs involved of 1 (range: 0 to 6). Tumor size ranged from 11 to 147 mm with a median of 43.5 mm; the median size of mediastinal adenopathies was 13 mm (0–78 mm).

### 3.2. Tissue and ctDNA Genotyping Results and ctDNA Test Performance

Tissue NGS detected 52 patients with gene alterations, ctDNA NGS detected 42 patients, corresponding to a detection rate of 45.2% and 36.5%, respectively. In total, 11 patients had concomitant alterations (co-mutations), with a median number of 1 (range: 1 to 3) alterations per patient, totalizing 66 hotspot mutations detected, 51 on both tissue and plasma, 12 exclusively on tissue, and 3 exclusively on plasma (Table 2 and Figure 1). *EGFR* and *KRAS* mutations were the most frequently found in both tissue and plasma samples (Figure 1). ctDNA allowed the detection of 3 additional oncogenic alterations not detected in the tissue sample, one KRAS c34G > T (Table 2, #39), one EGFR exon 19 deletion (Table 2, #43), and in an EGFR-mutated patient, a coexisting EGFR mutation was identified (Table 2, #93). In these three tissue negative cases, all samples were histological and corresponded to patients with stage IV disease. Combining tumor and plasma DNA analysis, 54 out of 115 (47%) patients presented hotspot alterations. A broad range of alterations was found, including mutations in *EGFR, KRAS, BRAF, HER2, ERBB4, ALK, TP53, STK11,* and *PIK3CA* (Figure 1).

For test performance calculation of the ctDNA targeted NGS assay, all alterations found and confirmed by dPCR were considered true-positive. Sensitivity was 81.0% (95% CI 69.1%–99.0%), specificity 95.3% (95% CI 86.9%–99.0%), NPV 83.6% (95% CI 72.3%–89.4%, PPV 94.4% (95% CI 75.3%–89.4%), and accuracy 88.2% (95% CI 81.3%–93.2%) (Table 3).

DNA variant allelic fraction was significantly higher in tissue (*p* < 0.001) with a median value of 27.0 (range: 0.2–96.5) for tissue NGS and 0.7 (range: 0.01–47.0) for plasma NGS (Figure 2).

### 3.3. Determinants of ctDNA Positivity

Clinical factors that could influence ctDNA positivity were investigated (Table 4; Appendix A). The presence of mutations on plasma, a surrogate of tumor circulating DNA, mostly occurred in patients with metastatic disease, when compared to those with locally advanced disease (*p* < 0.001), specifically with extra-thoracic location (*p* = 0.005), in those with multi-metastatic disease (*p* = 0.004), and was associated with a higher number of extra-thoracic organs involved (*p* = 0.013). Other factors, such as demographic features, performance status, paraneoplastic syndrome, comorbidities, tumor location, tumor size, and mediastinal lymph node involvement, did not influence ctDNA positivity. Following multivariate logistic analysis, only the number of organs involved (OR 1.428, 95% CI 1.055–1.932) was a ctDNA positivity-associated factor.

Regarding cell-free DNA (cfDNA) concentration, there were no significant differences between patients with (positive) or without (negative) detectable hotspot alterations. Cell-free-DNA concentrations among positive patients varying 0.10 ng/mL to 55.0 ng/mL (median: 1.10 ng/mL) and among negative patients from 0.13 ng/mL to 31.1 ng/mL (median: 0.42 ng/mL) (*p* = 0.093). Regarding the correlation between cfDNA amount and VAF, although a negative tendency (r = −0.003) appeared to be present, the correlation was not significant.

### 3.4. Correlation of ctDNA Positivity with Clinical Outcomes (PFS and OS)

For the entire cohort, progression-free survival to the first-line treatment (PFS1) achieved a median of 8 months (95% CI 6.29–9.70) and median OS of 12.0 months (95% CI 7.98–16.02), with a median follow-up time of 11 months (range: 0–74).

Based on tissue genotyping, EGFR mutated patients presented the highest median OS (23 months) and KRAS the worst (9 months). Additionally, TKIs-treated patients presented the highest median OS (23 vs. 10 months). The median PFS1 was almost identical (*p* = 0.505) in patients who were plasma positive and tissue positive, but median OS tended to be inferior (*p* = 0.067) to that obtained using tissue as the standard (Figure 3). Regarding cfDNA concentration, a negative correlation was found with the OS (r = −0.228, *p* = 0.019).

## 4. Discussion

Enhancing the detection of oncogenic alterations in LC is crucial to match patients to targeted therapies and clinical trials. The application of NGS platforms to LC tissue samples allowed the increase in the detection rate of druggable alterations, as previously demonstrated [7,8,16,17,18,19,20,21,22,23,24]. Still, this improvement does not resolve the limitation of tissue and DNA availability. Liquid biopsy for gene analysis can surmount this difficulty, contributing to the identification of driver mutations in newly diagnosed patients. The applicability of liquid biopsy to LC began with the analysis of EGFR mutational status on circulating cell-free DNA, as was documented in several studies [25,26] and meta-analysis [27,28]. Tests for plasma detection of EGFR mutations in the absence of a tumor sample are approved both by EMA and FDA, specifying the need to test the tissue if the result is negative due to the test’s suboptimal sensitivity. To overcome PCR-based assays associated limitations, particularly the limited range of genetic alterations comprised, NGS based plasma assays use is expanding. There is a considerable variation between platforms conferring differing sensitivity and concordance between plasma and tissue. In the pilot study of Conraud et al. with a multiplex PCR covering 12 different genomic regions, plasma had a sensitivity of 58% and a specificity of 87%, having tissue as the reference [29]. A bias-corrected targeted next-generation multiplexed cell-free DNA detected driver and resistance mutations with a sensitivity of 77% [30]. Additionally, Thompson et al., with a comprehensive genomic panel (Guardant 360) identified genomic alterations in 84% among driver resistant and potentially targetable alterations in patients with insufficient tissue sample or unable to pursue a biopsy [31]. BT Li et al., with a hybrid-capture 37 gene panel, found a sensitivity of 75% for oncogenic drivers in comparison to tissue genotyping results [32].

In our study, blood samples of 115 patients were compared with concurrent tissue samples. In both, an amplicon-based targeted gene platform, previously validated, was used. The ctDNA assay had an accuracy of 89.8%, sensitivity of 81.0%, specificity 95.3%, PPV 94.4%, and NPV of 83.6%. This data demonstrates the feasibility of using this NGS-based ctDNA to detect hotspot mutations due to its excellent test performance characteristics. Concordance was not perfect between plasma and tissue genotyping. The ctDNA based-NGS missed 12 out of 51 (23.5%) hotspot alterations detected in tissue. The lower rate of mutation detection in comparison to tissue is a limitation of ctDNA genotyping. These may be associated with a low concentration of tumor DNA on plasma, below the test’s detection limit. Additionally, lower tumor burden and some non-shedding tumors contribute to negative plasma results. The selected targeted-gene plasma assay had a smaller range of genes, but none of the missed alterations occurred in genomic regions not included in the panel. Therefore, negative results must be interpreted carefully, requiring confirmation with further tissue analysis. The imperfect concordance is not necessarily a handicap. ctDNA can find alterations missed on a tissue assay, as we described in 3 cases. Technical limitations of tissue assays and tumor heterogeneity are possible explanations. The presence of tumor DNA in plasma is a consequence of the shedding from multiple metastatic *foci* being more representative of tumor heterogeneity than a small tissue biopsy. The ability to detect additional cases with a ctDNA assay was also demonstrated in other studies. In the NILE study, a multicentric prospective study, the clinical utility of targeted ctDNA analysis to identify genomic alterations in untreated patients, including those with insufficient tissue, proved to increase the detection rate of targetable mutations more rapidly and effectively than tissue genotyping [33]. Aggarwal et al. achieved similar conclusions in their single center study [34], and the subgroup of patients who received targeted therapies based on plasma results achieved the expected response rate [34]. Both studies emphasize the integration of plasma NGS testing into the routine management of stage IV NSCLC due to increased detection of therapeutically targetable mutations. Our data also corroborate these results.

Looking at factors associated with the identification of mutations on ctDNA, we realized that the tumor burden was significantly associated with plasma positivity. Although most of the patients had stage IV disease, the detection of mutations was significantly higher in patients with metastatic disease than those with locally advanced disease, particularly those with extra-thoracic metastases and multiple organs involved. On the other hand, in cases without extra-thoracic disease and lower tumor burden, the change of absence of ctDNA is higher. We speculated that the presence of ctDNA at the time of initial diagnosis might have an adverse prognostic value, being associated with bulky disease. Similar data were found by Conrad et al. [29], the cfDNA concentration was associated with clinical stage and number of metastasis.

We explored the potential impact of incorporating ctDNA genotyping in the diagnostic workup by assessing correlations with clinical outcomes. We demonstrated that first-line treatment PFS was similar when the population was allocated according to the mutational status accessed by both assays, plasma and tissue. Looking exclusively to the subset of EGFR mutated population, PFS was highly concordant, corroborating the positive predictive value of NGS-based ctDNA assay to select patients for target treatment. In previous studies [25,35], the efficacy of 1st generation EGFR TKIs based on plasma assays was demonstrated. Further, Oxnard et al. and Remon et al. found that outcomes with osimertinib in patients with plasma T790M positivity were similar to patients positive by a tissue-based assay [12,36]. There was a discrepancy not statistically significant regarding OS, conferring lower OS to the plasma mutation-positive population and higher to the negative population due to the missed cases of targetable patients with the plasma assay. This reinforces the need for confirmation of negative results and is in line with results from other studies.

Targeted gene-panel has inherent drawbacks related to the selected primer designs, underestimating alterations in other genomic regions, which, ultimately, may not contribute to a comprehensive portrait of the real tumor biology. This panel is also limited to the detection of hotspot mutations in the coding regions of prespecified genes but unable to detect rearrangements. The constant increase in druggable alterations may imply panel modifications over time. Clinical validity was assessed on a retrospective basis, as ctDNA assay results were not provided in real-time, requiring prospective validation. Nevertheless, as already mentioned, PFS in the EGFR population, the most representative group of targetable alterations, was similar when patients were categorized based on plasma and tissue results.

Overall, besides these limitations, this amplicon-based targeted gene ctDNA assay demonstrated high accuracy, allowing the detection of a high percentage of patients with oncogenic alterations and some not detected on tissue NGS.

## 5. Conclusions

This study revealed the potential of integrating ctDNA analysis into the molecular LC diagnosis algorithm as a non-invasive test that contributes to identifying targetable genomic alterations and can helps guiding first-line therapy. Incorporating NGS with liquid biopsy is a crucial step forward, overcoming the limitation of tissue and DNA exiguity and avoiding the invasiveness associated with diagnostic techniques.

## Figures and Tables

**Figure 1 cancers-13-02707-f001:**
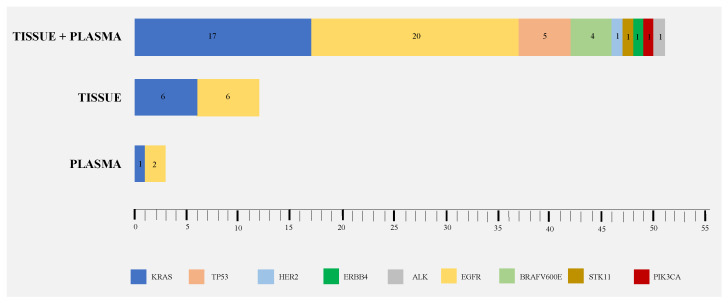
Number of mutations detected in tumor tissue and ctDNA NGS (66 hotspot mutations detected, 51 on both tissue and plasma, 12 exclusively on tissue, and 3 exclusively on plasma).

**Figure 2 cancers-13-02707-f002:**
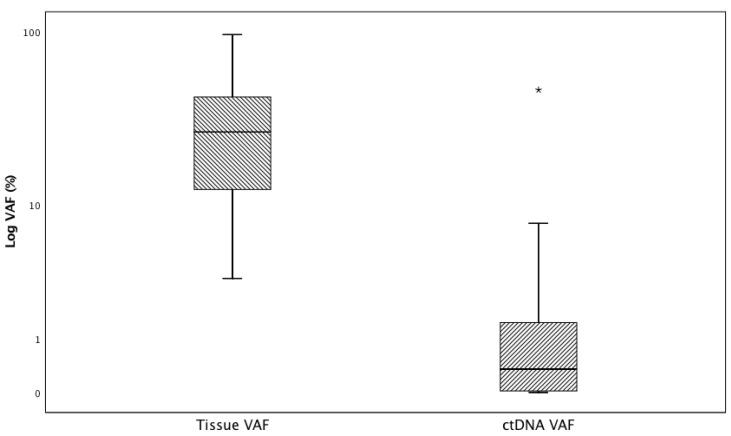
Variant Allelic Fraction of alterations found in tumor tissue and plasma ctDNA. * *p* < 0.001.

**Figure 3 cancers-13-02707-f003:**
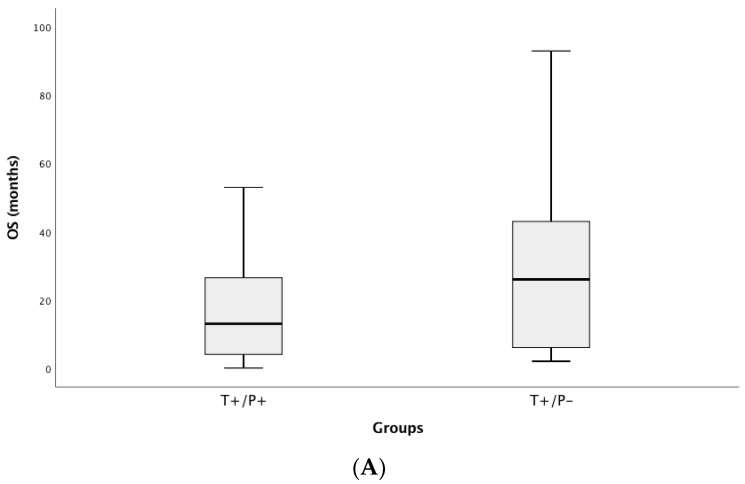
OS (**A**) and PFS (**B**) in tissue and plasma NGS positive and negative patients’ samples. Patients with mutations detected in both tissue (T+) and plasma (P+) present lower OS and PFS than those without mutations detected in plasma (P−) but present in tissue samples (T+/P−). PFS, progression-free survival to the first-line treatment; OS, overall survival. T, tissue; P, plasma.

**Table 1 cancers-13-02707-t001:** Patient’s demographics and clinical characteristics.

Characteristics		Value (*n*, %)
Age (median, range)		66 (38,92)
Gender, *n* (%)	Males	71 (61.7)
	Females	44 (38.3)
Performance Status, *n* (%)	0	42 (36.5)
	1	53 (46.1)
	2	15 (13.0)
	3	5 (4.4)
Smoking Status, *n* (%)	Smoker or Former smoker	73 (63.5)
	Never smoker	42 (36.5)
Histology, *n* (%)	Adenocarcinoma	114 (99.1)
	Adenosquamous	1 (0.9)
Tumor specimen type	Histologic	93 (83.9)
	Cytologic	22 (19.1)
Disease stage, *n* (%)	III: (IIIA/B/C)	21 (18.3): (7/10/4)
	IV	94 (81.7)
TNM discriminator, *n* (%)		
T	Tx	7 (6.1)
	T1	19 (16.5)
	T2	21 (18.3)
	T3	17 (14.8)
	T4	51 (44.3)
N	N0	43 (37.4)
	N1	10 (8.7)
	N2	36 (31.3)
	N3	26 (22.6)
M	M0	20 (17.4)
	M1a	22 (19.1)
	M1b	21 (18.3)
	M1c	52 (45.2)
First-Line treatment	BSC	9 (7.8)
	Multimodal (ChT+RT) ChT TKIs	13 (11.3) 69 (60.0) 24 (20.9)

**Table 2 cancers-13-02707-t002:** Comparison between ctDNA and tissue genotyping.

#	TNM	DNA Concentration ng/µL		Gene	Codon	Amino Acid	Variant Allelic Fraction %	
		Tissue	cfDNA				Tissue DNA	ctDNA
**Discordant Results (*n* = 15)**								
86	IIIB	36.80	0.41	KRAS	c.35G > T	p.(G12V)	44.70	0
3	IV	0.06	31.10	EGFR	c.2238_2252del15	p.(L747_T751del)	27.80	0
40	IV	1.55	0.44	EGFR	c.2235_2249del15	p.(E746_A750del)	56.50	0
29	IV	12.1	0.18	KRAS	c.35G > T	p.(G12V)	6.50	0
140	IV	2.83	2.44	EGFR	c.2239_2248del10insC	p.(L747_A750 > P)	14.00	0
54	IV	7.27	0.19	EGFR	c.2248_2276del29ins5	p.(A750_L760del)	84.50	0
23	IV	1.62	1.63	EGFR	c.2235_2249del15	p.(E746_A750del)	74.10	0
35	IV	1.17	0.25	KRAS	c.35G > T	p.(G12V)	3.30	0
37	IV	1.29	0.26	KRAS	c.34G > T	p.(G12C)	15.00	0
104	IV	0.11	1.00	KRAS	c.35G > T	p.(G12V)	50.60	0
118	IV	10.5	1.18	EGFR	c.2235_2249del15	p.(E746_A750del)	20.40	0
154	IV	30.8	0.13	KRAS	c.35G > T	p.(G12V)	46.30	0
39	IV	n.a.	3.25	KRAS	c.34G > T	p.(G12C)	0	0.45
43	IV	1.20	1.05	EGFR	c.2240_2257del18	p.(L747_P753 > S)	0	0.9
93 *	IV	5.70	0.76	EGFR	c.2235_2249del15	p.(E746_A750del)	0	0.11
**Concordant positive results (*n* = 51)**								
87	IIIB	4.14	0.27	KRAS	c.34G > T	p.(G12C)	5.90	0.21
163	IV	2.38	0.67	KRAS	c.35G > T	p.(G12V)	28.60	0.03
22	IV	0.22	0.44	EGFR	c.2294T > G	p.(V765G)	50.90	0.10
32	IV	2.58	0.27	KRAS	c.35G > T	p.(G12V)	19.90	0.12
62	IV	3.63	3.93	BRAF	c.1799T > A	p.(V600E)	36.30	0.53
81 *	IV	0.26	1.62	EGFR	c.2236_2250del15	p.(E746_A750del)	36.70	0.18
81 *				EGFR	c.2369C > T	p.(T790M)	15.00	0.14
93 *	IV	5.70	0.76	EGFR	c.2573T > G	p.(L785R)	31.30	0.18
146	IV	6.38	0.33	EGFR	c.2235_2249del	p.(E746_A750del)	35.80	1.39
100	IV	0.69	3.59	KRAS	c.35G > T	p.(G12V)	55.20	0.01
44 *	IV	n.a.	3.65	KRAS	c.38_39delGCinsAA	p.(G13G)	14.20	3.62
44 *				ERBB4	c.1033G > T	p.(A345S)	14.90	1.90
80 *	IV	4.17	1.55	BRAF	c.1799T > A	p.(V600E)	50.40	9.90
80 *				TP53	c.476C > G	p.(A159V)	39.20	4.90
119	IV	0.38	0.27	KRAS	c.34G > T	p.(G12C)	72.50	0.60
142	IV	22.0	n.a.	EGFR	c.2573T > G	p.(L785R)	15.30	0.12
165 *	IV	4.94	0.72	ERBB2	c.2310_2311insGCATAC	p.(A775_Gl776insT)	20.00	23.3
165 *				TP53	c.1024C > T	p.(R342 *)	21.00	1.00
187	IV	n.a.	0.10	EGFR	c.2235_2249del15	p.(E746_A750del)	31.70	9.05
1	IV	2.53	1.31	KRAS	c.35G > A	p.(G12A)	6.10	1.37
2	IV	0.34	3.36	EGFR	c.2240_2257del18	p.(L747_P753 > S)	40.00	0.72
12	IV	1.35	6.38	TP53	c.527G > T	p.(C176F)	41.10	1.27
14	IV	2.45	55.00	KRAS	c.35G > T	p.(G12V)	11.70	0.09
15 *	IV	0.06	2.27	KRAS	c.182A > G	p.(Q61R)	8.80	0.43
15 *				TP53	c.461G > T	p.(Gl154V)	24.70	0.71
15 *				STK11	c.597G > C	p.(E199D)	12.80	0.53
16	IV	0.18	1.01	KRAS	c.34G > T	p.(G12C)	3.80	0.72
38	IV	0.95	0.19	EGFR	c.2573T > G	p.(L785R)	6.30	0.91
52	IV	2.95	0.82	EGFR	c.2236_2250del15	p.(E746_A750del)	0.17	4.10
51	IV	n.a.	3.91	KRAS	c.34G > T	p.(G12C)	5.90	1.50
57	IV	3.39	2.67	EGFR	c.2573T > G	p.(L785R)	29.20	0.03
60	IV	4.46	1.26	KRAS	c.35G > T	p.(G12V)	20.40	7.72
65	IV	0.28	0.28	EGFR	c.2235_2249del15	p.(E746_A750del)	7.60	2.01
74 *	IV	11.70	17.7	EGFR	c.2236_2250del15	p.(E746_A750del)	11.10	0.27
74 *				KRAS	c.182A > G	p.(G61R)	0.38	0.07
98 *	IV	0.96	1.26	EGFR	c.2239_2248del	p.(L747_A750 > P)	66.00	0.22
98 *				ALK	c.3512T > A	p.(I1171N)	0.08	0.08
95	IV	n.a.	0.55	EGFR	c.2240_2254del15	p.(L747_T751del)	96.50	0.65
101	IV	0.39	1.29	KRAS	c.34G > T	p.(G12C)	17.40	0.06
107 *	IV	22.10	1.03	EGFR	c.2240_2257del18	p.(L747_P753 > S)	67.20	5.50
107 *				EGFR	c.2369C > T	p.(T790M)	0.60	1.25
117	IV	48.80	0.39	KRAS	c.35G > A	p.(G12A)	22.30	47.0
130	IV	n.a.	0.56	EGFR	c.2235_2249del15	p.(E746_A750del)	13.20	1.06
131	IV	0.93	n.a.	BRAF	c.1799T > A	p.(V600E)	29.10	0.61
144 *	IV	0.51	0.37	EGFR	c.2235_2249del15	p.(E746_A750del)	34.60	0.62
144 *				PIK3CA	c.1633G > A	p.E545K)	27.80	0.75
143	IV	0.55	2.10	BRAF	c.1799T > A	p.V600E)	10.90	1.89
136 *	IV	8.85	1.14	KRAS	c.35G > T	p.(G12V)	26.70	10.30
136 *				TP53	c.839G > A	p.(R280K)	13.00	0.06
161	IV	13.90	0.58	KRAS	c.35G > T	p.(G12V)	27.40	2.22
78	IV	2.15	2.53	EGFR	c.2573T > G	p.(L785R)	18.60	0.09
**Concordant negative results (*n* = 61)**								

(cfDNA, cell-free DNA; ctDNA, circulating tumor DNA). In this cohort: 15 discordant alterations; 51 concordant positive hotspot alterations; the remaining 61 cases presented no alterations both on tissue and plasma; * cases with co-mutations.

**Table 3 cancers-13-02707-t003:** Analysis of concordance between plasma and tissue genotyping on a patient basis (a), and hotspot alterations basis (b).

Cases Compared (N)	Concordant Cases [tDNA vs. ctDNA (N)]		Discordant Cases [tDNA vs. ctDNA (N)]		Concordant Cases (%)	Kappa
	Negative/Negative	Positive/Positive	Negative/Positive	Positive/Negative		
(a) 115 patients	61	40	2	12	87.8	0.75
(b) 127 hotspot	61	51	3	12	88.2	0.76

**Table 4 cancers-13-02707-t004:** Clinical factors determining ctDNA positivity.

TNM Discriminator Value, *n* (%)		All	Plasma-Negative	Plasma-Positive	*p* Value
Age (median, IQR)		66 (14)	66 (12)	64 (18)	0.716
Gender	Male	71 (61.7)	46 (63.0)	25 (59.5)	0.711
	Female	44 (38.3)	27 (37.0)	17 (40.5)	
Performance Status	0	42 (36.5)	35 (47.9)	7 (16.7)	**0.009**
	1	53 (46.1)	29 (39.7)	25 (59.5)	
	2	15 (13.0)	7 (9.6)	8 (19.0)	
	3	5 (4.4)	2 (2.8)	2 (4.8)	
TNM discriminator					
T, *n* (%)	Tx	7 (6.1)	2 (2.7)	5 (11.9)	0.328
	T1	13 (11.3)	7 (9.6)	6 (14.3)	
	T2	15 (13.0)	10 (13.7)	5 (11.9)	
	T3	12 (10.4)	7 (9.6)	5 (11.9)	
	T4	68 (59.1)	47 (69.1)	21 (50.0)	
N, *n* (%)	N0	4 (38.3)	26 (35.6)	18 (42.9)	0.623
	N1	10 (8.7)	7 (9.6)	3 (7.1)	
	N2	35 (30.4)	21 (28.8)	14 (33.3)	
	N3	26 (22.6)	19 (26.0)	7 (16.7)	
M, *n* (%)	M0	20 (17.4)	19 (26.0)	1 (2.4)	**0.013**
	M1a	22 (19.1)	13 (17.8)	9 (21.4)	
	M1b	21 (18.3)	13 (17.8)	8 (19.0)	
	M1c	52 (45.2)	28 (38.4)	24 (57.1)	
Number of organs involved Median (min-max)		1 (0–6)	1 (0–6)	2 (0–5)	**0.004**
Disease stage	IIIA	7 (6.1)	7 (9.6)	0	**0.010**
	IIIB	10 (8.7)	9 (12.3)	1 (2.4)	
	IIIC	4 (3.5)	4 (3.5)	0	
	IV	94 (81.7)	53 (72.6)	41 (97.6)	

Bold: *p* < 0.05.

## Data Availability

The data presented in this study are available in Appendix A.

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
