# Peer review of "Clinical Application of Next-Generation Sequencing of Plasma Cell-Free DNA for Genotyping Untreated Advanced Non-Small Cell Lung Cancer"

_cancers, 2021, doi:10.3390/cancers13112707_

Round 1

Reviewer 1 Report

Like many articles on the subject, this work highlights the interest of molecular characterization of circulating DNA in the management of cancer in general and lung cancer in particular.

For this the authors use the targeted NGS oncomine lung cfDNA assay for plasma and the ion ampliseq colon and lung cancer research panel V2 for biopsies.

It would be necessary to specify the analysis pipeline of the sequence data. But also to indicate the parameters retained for sequencing: minimum number of total reads, number of reads retained to validate a variant for each of the two analyses performed. The authors should specify the respective sensitivity of each technique used. This is an important point for comparison since the cf DNA lung assay uses unique molecular barcode

The authors should explain  why they did not use the same technique and the more sensible  for both biological sample sources ?

The authors verify the double negatives by digital PCR. What is the threshold sensitivity of the test performed which is retained for the confirmation of the NGS result.

In Table 2 the three discordant samples 39, 43 and 93 are intriguing, was the negativity of the tissues checked by ddPCR, if so what are the results?

The authors indicate that tumor size does not positively influence the result obtained on circulating DNA. It would be useful for readers to show tumor volumes, localization and allelic frequencies in a table.

The authors indicate that the detection of mutations is significantly higher in metastatic patients, but this is not always true for cerebral metastasis. In this case a search for tumor DNA can be performed in the cerebrospinal fluid.

Author Response

Like many articles on the subject, this work highlights the interest of molecular characterization of circulating DNA in the management of cancer in general and lung cancer in particular.

For this the authors use the targeted NGS oncomine lung cfDNA assay for plasma and the ion ampliseq colon and lung cancer research panel V2 for biopsies.

It would be necessary to specify the analysis pipeline of the sequence data. But also to indicate the parameters retained for sequencing: minimum number of total reads, number of reads retained to validate a variant for each of the two analyses performed. The authors should specify the respective sensitivity of each technique used. This is an important point for comparison since the cf DNA lung assay uses unique molecular barcode

Answer: We thank the reviewer for the observation. We completely agree with the importance of such information and the impact it has on the quality of the end results. Precisely due to that, the pipelines and workflows used were the ones provided by the manufacturer Thermo Fisher Scientific for each of the assays used. No changes to the default settings were done. We have included the different versions of the bioinformatic pipelines used (Torrent Suite Software v.5.2 and Ion Reporter versions 5.6 and 5.10 (Thermo Fisher Scientific). Concerning to the sensitivity for the techniques these are commercially available solutions. According to the manufacturer the Oncomine Lung cfDNA Assay has a detection limit of 0.1% frequency (with a sensitivity of >90% and specificity of >98%), or 1 mutant copy in a background of 1,000 wild-type (WT) copies. To achieve a 0.1% limit of detection (LOD), 20 ng of input cfDNA is required. Lower amounts of cfDNA can be used, but the limit of detection may be higher depending on the input amount. This information is detailed in the supplementary data.

The authors should explain why they did not use the same technique and the more sensible for both biological sample source?

Answer: Thank you. This project was developed over several years. Due to the limitations on biological material obtained from each patient, for the molecular characterization of each sample, it was used the method routinely applied at the time. This further strengthens the power of the study.

The authors verify the double negatives by digital PCR. What is the threshold sensitivity of the test performed which is retained for the confirmation of the NGS result.

Answer: The sensitivity for variant detection, according to the manufacturer, is down to 0.1%. This information was added to the supplementary data.

In Table 2 the three discordant samples 39, 43 and 93 are intriguing, was the negativity of the tissues checked by ddPCR, if so what are the results?

Answer: Thank you for the comment. We agree with the reviewer observation. Whenever there was available biological material, digital PCR was used to confirm the results. Samples were tested by digital PCR and confirm the negative results obtained by NGS.

The authors indicate that tumor size does not positively influence the result obtained on circulating DNA. It would be useful for readers to show tumor volumes, localisation and allelic frequencies in a table.

Answer: Thank you for the constructive suggestion. We agree with the reviewer. A new table with the requested information was added (supplementary data 5).

Plasma-positive

Plasma-negative

P value

Predominant Location, n (%)

Lower lobes

Upper lobes

Bilateral

Hemithorax

18 (43.9)

17 (41.5)

3 (7.3)

3 (7.3)

28 (38.9)

35 (48.6)

6 (8.3)

3 (4.2)

0.808

Size (T), median (min-max)

47.0 (11.0-147.0)

40.5 (15.0-112.0)

0.417

Size (N), median (min-max)

14.0 (0-78.0)

13.0 (0-60.0)

0.381

Tumor volume, median (min-max)

21203.0 (445-556321.5)

19654.6 (648-475154.5)

0.879

Type of sample, n (%)

Cytologic

Histologic

7 (16.7)

35 (83.3)

15 (20.5)

58 (79.5)

0.610

The authors indicate that the detection of mutations is significantly higher in metastatic patients, but this is not always true for cerebral metastasis. In this case a search for tumor DNA can be performed in the cerebrospinal fluid.

Answer: Thanks for the comment. Patients with brain tumours or exclusively brain metastasis do not present with or present with low amounts of ctDNA in plasma, precluding the genomic characterization through plasma ctDNA. In these cases, ctDNA can be more abundantly present in the cerebrospinal fluid (CSF) than in plasma, and a liquid biopsy of CSF would be a good option, as suggested. In our study, seventeen patients had documented brain metastasis at the inclusion of the study, but only 2 were oligometastatic. Regarding ctDNA results, 7 had a mutation detected on cfDNA, and 10 had no mutation detected.  It is not possible to assume definite conclusions regarding this topic.

Reviewer 2 Report

General comment: Fernandes and colleagues studied the clinical value of an DNA-based NGS for detecting actionable mutations in plasma in patients with advanced lung adenocarcinoma.

Biomarker testing is necessary to determine the optimal treatment of newly diagnosed patients with advanced NSCLC, and guidelines recommend not only testing for EGFR and BRAF, but also for identifying other relevant biomarkers with approved drugs in the first-line setting.

The main conclusion of the authors is that the NGS test used may directly guide first-line therapy.

I would like the authors to justify or comment on this fact. What strategy will the authors recommend to identify all the alterations recommended by the different guidelines in advanced NSCLC. For example, will they use an alternative method for gene fusions, will they perform these tests on tissue or other?

  • Page 1. Line23. Please specify patients are advanced NSCLC.
  • Page 1 & 3. Please specify if this study is prospective or retrospective.
  • Page 2. Line 79. “have low DNA content”– presumably this means that have scarce material or scarce tumor sample.
  • Page 2. Line 91. “low DNA content”. Do you mean low tumor DNA content?
  • Page 3, last paragraph. Please specify which patients were included in the study -advanced NSCLC.
  • Page 3, line 113. Specify the time period for patients recruitment.
  • Page 3, line 125. Figure S1. Please specify the n performed by NGS ctDNA and tissue samples.
  • Page 3, line 130. Replace “tumor” with “tissue”.
  • Please italicize all gene names.
  • Page 3, line 130. Specify that the NGS cfDNA assay can only detect SNV and short indels.
  • Page 4 Line154. The authors comment that ‘Biopsy and cytology specimens were reviewed by a pathologist’ but in the text and supplementary figure it is not mentioned that cytology samples were used for the analysis. Please correct.
  • Page 4 Line166. From the text: “Libraries were generated using 1–10 ng”. Does it mean that 1ng is the minimum DNA content measured before attempting sequencing?
  • Please add tissue quantity specifications for molecular analysis (number of slides, thickness, minimum tumor area, % tumor infiltration). Add the same information if cytological samples were included in the study (minimum of tumor cells to consider a valid sample for DNA isolation).
  • What is the detection limit for NGS-based plasma genotyping?
  • Add in figure 1. What represent the numbers, the N of cases mutated or the %? Please specify. Also, it would be nice to add this information within the text, at least which alterations were the most frequent identified.
  • Table 2. Could it be that these discordant samples are due to stage (not IV stage) or because the patient do not have an extrathoracic or multi-metastatis disease? The comparison is only between tissue and plasma or also with cytology and plasma?
  • Page 10. What was the median follow-up of the patients?
  • Add the meaning of T and P in figure legend 3. What is NSG?
  • Page 13. Line 347. Add % (12/51).

Author Response

General comment: Fernandes and colleagues studied the clinical value of an DNA-based NGS for detecting actionable mutations in plasma in patients with advanced lung adenocarcinoma. Biomarker testing is necessary to determine the optimal treatment of newly diagnosed patients with advanced NSCLC, and guidelines recommend not only testing for EGFR and BRAF, but also for identifying other relevant biomarkers with approved drugs in the first-line setting. The main conclusion of the authors is that the NGS test used may directly guide first-line therapy.

I would like the authors to justify or comment on this fact. What strategy will the authors recommend to identify all the alterations recommended by the different guidelines in advanced NSCLC. For example, will they use an alternative method for gene fusions, will they perform these tests on tissue or other?

Answer: Thank you for the question. Our results indicate that a ctDNA‐NGS based test has high accuracy and a high concordance between positive cases. Also, ctDNA-NGS identified additional patients with actionable genomic alterations and could, therefore, be used to complement traditional tissue‐based testing for NSCLC. It could be added to tissue testing when tissue DNA is not enough for sequencing, especially in high-risk patients for biopsies and in those with no detectable druggable alterations found in the tissue. Considering the hypothetical faster turn-around time and less cost, it could be the front-line test for genotyping, though this was not evaluated in this study, and authors cannot state it. It is a crucial point to be scrutinised in a future study.

Regarding the spectrum of alterations that must be detected, and are recommended at the present time, this panel is of limited value to detect fusions, as mentioned in the discussion. Due to the temporal nature of the study, only recently analytical tools were developed for fusion detection in cfRNA. Currently, we are performing a similar study to evaluate the capacity to use cfRNA for detection of fusions in NSCLC patients.

To overcome this limitation, a DNA+RNA panel combined panel can be used. An alternative is a stepwise approach. If no alteration is found, a panel able to detect fusions can be used subsequently.

Page 1. Line 23. Please specify patients are advanced NSCLC.

Answer: Changed accordingly

Page 1 & 3. Please specify if this study is prospective or retrospective.

Answer: Prospective. This information was included in the text.

Page 2. Line 79. “have low DNA content”– presumably this means that have scarce material or scarce tumor sample.

Answer: Thanks for the correction. It was included in the text.

Page 2. Line 91. “low DNA content”. Do you mean low tumor DNA content?

Answer: Thanks for the correction. It was included in the text.

Page 3, last paragraph. Please specify which patients were included in the study -advanced NSCLC.

Answer: Thanks for the comment. It was clarified in the text.

Page 3, line 113. Specify the time period for patients recruitment.

Answer: Thanks for the comment. It was clarified in the text.

Page 3, line 125. Figure S1. Please specify the n performed by NGS ctDNA and tissue samples.

Answer: Thanks for the comment. It was corrected in the S1 figure of the supplementary data.

Page 3, line 130. Replace “tumor” with “tissue”.

Answer: Done

Please italicize all gene names.

Answer: Done

Page 3, line 130. Specify that the NGS cfDNA assay can only detect SNV and short indels.

Answer: Thanks for the correction. It was clarified in the text.

Page 4 Line 154. The authors comment that ‘Biopsy and cytology specimens were reviewed by a pathologist’ but in the text and supplementary figure it is not mentioned that cytology samples were used for the analysis. Please correct.

Answer: Thanks for the comment. Information regarding that topic was included in the text and better discriminate in the supplementary data.

Page 4 Line166. From the text: “Libraries were generated using 1–10 ng”. Does it mean that 1ng is the minimum DNA content measured before attempting sequencing?

Answer: In some of the samples the amount was so minute that the attempts to measure were unsuccessful. In these, in order not to spend all the remaining biological material for quantification, NGS libraries were attempted independently of quantification.

Please add tissue quantity specifications for molecular analysis (number of slides, thickness, minimum tumor area, % tumor infiltration). Add the same information if cytological samples were included in the study (minimum of tumor cells to consider a valid sample for DNA isolation).

Answer: For molecular analysis all samples were reviewed by trained pathologists. 4-5 FFPE tissue sections of 10 µm thickness with at least 10% tumor cells were used to isolate DNA. All genetic analyses were done at IPATIMUP, a College of American Pathologists and ISO15189 accredited laboratory. This information was added to the text.

What is the detection limit for NGS-based plasma genotyping?

Answer: Thank you for the question. The detection limit of the assay for plasma genotyping is dependent on the initial amount of cfDNA used for library preparation. According to the manufacturer with 20 ng cfDNA it reaches a sensitivity of 0.1% AF.

Add in figure 1. What represent the numbers, the N of cases mutated or the %? Please specify. Also, it would be nice to add this information within the text, at least which alterations were the most frequent identified.

Answer: The numbers represent the “n” of mutated cases (figure legend was changed accordingly). The text related to the figure is mentioned in the first paragraph of point 3.2.

Table 2. Could it be that these discordant samples are due to stage (not IV stage) or because the patient do not have an extrathoracic or multi-metastatis disease? The comparison is only between tissue and plasma or also with cytology and plasma?

Answer: Discordant cases (Tumor negative/ctDNA positive) corresponded to stage IV disease (T4N2M1a; T4N0M1c; T3N0M1a) and all were histological samples (2 lung tissue obtained by transthoracic core needle biopsy and 1 bronchial biopsy by bronchoscopy). This information was briefly included in the text.

Page 10. What was the median follow-up of the patients?

Answer: Data censoring was done when 90% of the patients had deceived, corresponding to a median follow-up time was 11 months (range 0 to 74), which reinforces the robustness of survival analysis. This data was included in the text.

Add the meaning of T and P in figure legend 3. What is NSG?

Answer: Added the requested information in figure legend. NSG is a typo, we changed to NGS.

Page 13. Line 347. Add % (12/51).

Answer: Thanks. The percentage (23.5%) was added.

Round 2

Reviewer 1 Report

All my concerns and comments were satisfactorily addressed.

Reviewer 2 Report

The authors have addressed all my concerns.